# Relationship between Baseline Serum Potassium and 1-Year Readmission in Pediatric Patients with Heart Failure: A Retrospective Cohort Study

**DOI:** 10.3390/children11060725

**Published:** 2024-06-14

**Authors:** Yong Han, Yuqin Huang, Danyan Su, Dongli Liu, Cheng Chen, Yusheng Pang

**Affiliations:** Department of Pediatrics, The First Affiliated Hospital of Guangxi Medical University, No 6 Shuangyong Road, Nanning 530021, China; yoyohan1234@sr.gxmu.edu.cn (Y.H.); yfy107393@sr.gxmu.edu.cn (Y.H.); sudanyeal@sr.gxmu.edu.cn (D.S.); dongli.liu@sr.gxmu.edu.cn (D.L.); yfy004118@sr.gxmu.edu.cn (C.C.)

**Keywords:** pediatric heart failure, serum potassium, readmission, optimal range, prognosis

## Abstract

Pediatric heart failure (HF) is associated with high readmission rates, but the optimal serum potassium range for this population remains unclear. In this single-center retrospective cohort study, 180 pediatric patients hospitalized for HF between January 2016 and January 2022 were stratified into low-potassium (<3.7 mmol/L), middle-potassium (3.7–4.7 mmol/L), and high-potassium (≥4.7 mmol/L) groups based on the distribution of potassium levels in the study population. The primary outcome was readmission for HF within 1 year of discharge. Cox regression and restricted cubic spline models were used to assess the association between potassium levels and 1-year HF readmission rates. Notably, 38.9% of patients underwent 1 or more 1-year readmissions for HF within 1 year. The high-potassium group had a significantly higher readmission frequency than the middle-potassium group. In multivariate Cox regression models, potassium levels of ≥4.7 mmol/L were independently associated with increased 1-year readmission risk. A J-shaped relationship was observed between baseline potassium levels and 1-year readmission risk, with the lowest risk at 4.1 mmol/L. In pediatric patients with HF, a serum potassium level ≥ 4.7 mmol/L was independently associated with increased 1-year readmission risk. Maintaining potassium levels within a narrow range may improve outcomes in this population.

## 1. Introduction

Pediatric heart failure (HF) is a growing concern, with an increasing prevalence and substantial morbidity and mortality [1,2]. Despite advances in medical management, HF remains the leading cause of hospitalization among children with cardiovascular diseases, and readmission rates within 1–2 years after diagnosis remain high [3,4,5]. The complex pathophysiology of HF in children, coupled with the unique challenges of pediatric pharmacotherapy, necessitates a deeper understanding of the factors that influence outcomes in this vulnerable population [6,7,8].

Electrolyte imbalance, particularly dyskalemia, is common in patients with HF and has been associated with adverse outcomes [9,10]. However, the optimal serum potassium range for pediatric patients with HF remains poorly defined. In healthy children, the normal potassium range is typically 3.5–5.5 mmol/L; however, this may not be applicable to those with HF. Studies in adults have suggested that a narrower potassium range of 4–5 mmol/L is associated with improved outcomes in HF [11,12]; however, similar data in pediatric cohorts are lacking.

Given the paucity of evidence on potassium management in children with HF, there is a critical need to elucidate the relationship between serum potassium levels and clinical outcomes in this population. Identifying the optimal potassium range associated with a reduced readmission risk could inform clinical decision-making and improve patient care. In this study, we aimed to identify the optimal serum potassium range associated with the lowest risk of 1-year readmission in pediatric patients with HF.

## 2. Materials and Methods

### 2.1. Study Population

We conducted a single-center retrospective analysis of all pediatric patients aged <18 years hospitalized for HF between January 2016 and January 2022. Patients who fulfilled both the criteria for symptomatic HF by presenting with signs or symptoms (tachypnea, diaphoresis, hepatomegaly, growth retardation, fatigue, peripheral edema, or ascites) or echocardiographic features of left ventricular systolic dysfunction (left ventricular ejection fraction [LVEF] < 55%) were considered eligible [13]. Due to the immature metabolism and renal function of newborns, infants aged <1 month were excluded from the study. We focused on HF attributable to ventricular dysfunction and excluded patients with HF caused by intracardiac shunts or left-sided obstructive lesions. Patients who died during index hospitalization and those who underwent hemodialysis were also excluded. Additionally, patients without baseline potassium measurements or 1-year follow-up data were excluded from the study. Finally, our study included 180 pediatric patients aged <18 years who were hospitalized for HF and met the inclusion criteria (Figure 1). This study was approved by the research ethics committee of our institution (approval number: 2023-E478-01) and was conducted in accordance with the principles outlined in the Declaration of Helsinki.

### 2.2. Clinical Data and Laboratory Values

Comprehensive baseline data were obtained from initial hospitalization records, including (1) clinical data, such as sex, age, ethnicity, HF etiology, prior history of HF, New York Heart Association (NYHA)/Ross classification at admission, and length of hospital stay; (2) laboratory indicators and echocardiographic findings (Table 1); and (3) discharge medications, including renin–angiotensin–aldosterone system inhibitors (RAASi), digoxin, diuretics, and β-blockers. For patients with multiple admissions, only the data from the first admission were included to minimize confounding factors. Laboratory data were collected within 24 h of admission, whereas echocardiography and electrocardiography data were obtained within 48 h. Prior history of HF was defined as having a medical history of HF for more than 6 months prior to this study. The estimated glomerular filtration rate (eGFR) was calculated using the combined creatinine–cystatin C-based 2012 CKiD equation, and LVEF was assessed from the parasternal long-axis view using the Teichholz method.

### 2.3. Study Outcome

All patients were followed up through telephonic interviews with caregivers and a review of hospital records. The endpoint of the study was readmission for HF within 1 year of discharge.

### 2.4. Statistical Analysis

To ensure sufficient statistical power, the serum potassium cutoff value was determined based on the distribution of potassium levels in the study population. Specifically, a Cox inflection point analysis was employed to identify a more precise cutoff value for this study, revealing a narrower-than-usual range for serum potassium (Break point 1 = 3.7 and Break point 2 = 4.7) (Appendix A). Patients were stratified into three groups according to serum potassium levels using cutoffs from the Cox inflection point analysis as follows: the low-potassium group (<3.7 mmol/L), the middle-potassium group (3.7−4.7 mmol/L), and the high-potassium group (≥4.7 mmol/L). Differences among the three groups were compared using the following statistical tests based on the type of variable: chi-square tests for categorical variables, one-way analysis of variance for variables with a normal distribution, and Kruskal–Wallis tests for variables with a skewed distribution.

Bar charts were used to illustrate the differences in the incidence of 1-year readmission among patients with varying serum potassium levels. Time-to-event data were presented using Kaplan–Meier cumulative incidence curves with log-rank tests. Although death before readmission was treated as a competing risk for the readmission endpoint, the small number of competing events (*n* = 6) may have limited the ability to detect a significant competing risk between death and readmission in this study.

Cox regression and restricted cubic spline (RCS) models were used to assess the association between potassium levels and 1-year HF readmission. The covariates for inclusion in the regression model were selected on the basis of their clinical relevance, univariate analysis results, and the principle of selecting variables whose matched odds ratios changed by at least 10% after inclusion in the model. Prior to the multivariate analysis, significant univariate variables were assessed for collinearity using a tolerance cutoff of <0.02 and a variance inflation factor cutoff of >5 (Appendix A). Variables without collinearity were included in the multiple regression model and presented as hazard ratios (HRs) with 95% confidence intervals (CIs) and corresponding *p* values. Three multivariate Cox regression models were adjusted for different covariates as follows: Model 1 was adjusted for age and sex; Model 2 was adjusted for the covariates included in Model 1 as well as serum creatinine (SCr), blood urea nitrogen (BUN), serum uric acid (SUA), and eGFR; and model 3 was adjusted for the covariates included in Model 2 in addition to LVEF, NYHA/Ross class, and serum albumin (ALB) and sodium levels. In multivariate Cox regression models, potassium levels were incorporated as both continuous and categorical variables to assess their impact on the outcome of interest. Furthermore, we determined the other risk factors for 1-year HF readmission using multivariate Cox regression analysis. Variables with more than 10% missing values were excluded, whereas those with <10% missing values were computed using mean imputation.

Continuous variables with normal distributions were expressed as mean ± standard deviation, while continuous variables with skewed distributions were expressed as median [interquartile range (IQR)]. Categorical variables were presented as n (%). All statistical tests were two-sided, and statistical significance was set at *p* < 0.05. All statistical analyses were performed using SPSS (version 27.0; IBM Inc., New York, NY, USA) and R Statistical Software version 4.2.0 (R Foundation (n.d.). The R Project for Statistical Computing. Vienna, Austria. http://www.R-project.org).

## 3. Results

### 3.1. Baseline Characteristics of the Patients

In this study, we analyzed data from 180 patients with a mean age of 7.0 ± 5.6 years. Among them, 106 (58.9%) were male and 74 (41.1%) were female patients. A total of 104 (57.8%) patients had NYHA/Ross class III or IV symptoms, and 30 (16.7%) had a history of HF more than 6 months prior to current admission. Patients in the high-potassium group demonstrated a greater prevalence of preexisting HF, worse NYHA/Ross class, and diminished LVEF and eGFR at admission (*p* < 0.05). The types of medications prescribed at discharge were similar among the three groups. Digoxins, diuretics (hydrochlorothiazide and spironolactone), and RAASi (captopril) were administered most frequently. However, β-blockers (metoprolol) were prescribed after discharge in most cases. Table 1 presents the baseline characteristics of the patients and their associations with the different potassium groups.

### 3.2. Distribution of Baseline Potassium Levels

Among the analyzed patients, 27 (15%), 105 (58.3%), and 48 (26.7%) were classified into the low-potassium (<3.7 mmol/L), middle-potassium (3.7–4.7 mmol/L), and high-potassium group (≥4.7 mmol/L), respectively. As illustrated in Figure 2A, the majority of patients had potassium levels within the normal reference range (K^+^ 3.5–5.5 mmol/L, 84.4%). As illustrated in Figure 2B, higher potassium levels were more prevalent in patients with lower eGFR.

### 3.3. Endpoint Event Rates Stratified by Baseline Potassium Levels

Among the study population, 38.9% experienced at least one readmission for HF within 1 year, with 17.8% having two or more readmissions. The mean number of readmissions per patient readmitted at least once was 2.1 ± 1.3 (median, 1.2; IQR, 1–3). Patients in the high-potassium group exhibited a significantly higher frequency of readmission than that of patients in the middle-potassium group (54 vs. 29.5%, *p* = 0.008; Figure 3A). The Kaplan–Meier estimates at 1 year revealed a markedly increased incidence of HF-related readmissions in patients in the high-potassium group compared to those in the low- and middle-potassium groups (Figure 3B).

### 3.4. Association between Baseline Potassium Levels and Outcomes

Three multivariate Cox regression models were developed to investigate the role of serum potassium level as an independent predictor of 1-year HF readmission (Table 2). In the unadjusted model, a 1 mmol/L increase in serum potassium levels was associated with a 1.85-fold higher risk of 1-year readmission (95% CI 1.19–2.87, *p* = 0.006). This association remained significant after adjusting for age and sex but was attenuated after further adjustments for renal function indices (BUN, SCr, SUA, and eGFR). In addition, we converted serum potassium levels from a continuous variable to a categorical variable. Using the middle-potassium group as the reference, the high-potassium group was associated with a higher incidence of 1-year readmission in all models (Model 1: HR = 2.02, 95% CI 1.16–3.53, *p* = 0.013; Model 2: HR = 2.03, 95% CI 1.17–3.51, *p* = 0.012; model 3: HR = 1.91, 95% CI 1.09–3.35, *p* = 0.024). No significant differences were observed in the 1-year readmission rates between the low and middle-potassium groups in any model.

The RCS curve illustrates a J-shaped relationship between 1-year readmission risk and baseline potassium levels (Figure 4). The lowest risk was observed at 4.1 mmol/L.

### 3.5. Other Risk Factors for 1-Year HF Readmission in Pediatric Patients with HF

Univariate and multivariable Cox regression analyses of the risk predictors of 1-year HF readmission in pediatric patients with HF are shown in Table 3. Ultimately, through multivariate analysis, we found that BUN levels (HR = 1.09, 95% CI 1.01–1.18), NYHA/Ross class III or IV (HR = 1.94, 95% CI 1.09–3.46), and potassium levels ≥ 4.7 mmol/L (HR = 1.87, 95% CI 1.08–3.26) were independent risk factors for 1-year readmission in this cohort (all *p* < 0.05).

## 4. Discussion

In our cohort of pediatric patients with HF, we observed that pediatric patients with potassium levels of >4.7 mmol/L had a higher risk of HF readmission within 1 year, even when potassium levels were within the normal reference range. Our findings suggest that a baseline serum potassium level of ≥4.7 mmol/L is an independent predictor of 1-year HF readmission in pediatric patients with HF. Furthermore, we identified a J-shaped association between baseline potassium levels and the risk of 1-year readmission in pediatric patients with HF. The optimal potassium level associated with the lowest risk was 4.1 mmol/L. To our best knowledge, this is the first study to demonstrate that a potassium level of ≥4.7 mmol/L is an independent risk factor for 1-year HF readmission in the pediatric population with HF.

Our results are consistent with those of recent studies on adult HF populations. Toledo et al. found that potassium levels of ≥4.7 mmol/L may facilitate the identification of patients with HF at an increased risk of adverse outcomes [14]. Aldahl et al. reported that the optimal potassium range for patients with chronic HF is 4.1–4.8 mmol/L, and deviations from this range are associated with increased short-term mortality risk [9]. Furthermore, Shang et al. reported that potassium levels of ≥4.7 mmol/L are associated with 1-year readmission in older patients with HF [15]. While the aforementioned studies provide valuable insights into the relationship between serum potassium levels and outcomes in adult patients with HF, there is a paucity of data on this association in the pediatric population with HF. Lin et al. found that electrolyte abnormalities, including hypokalemia and hyperkalemia, are associated with increased mortality in pediatric intensive care units; however, the study did not specifically examine the optimal potassium range [16]. Another study by Maryam et al. reported that hyperkalemia is common in pediatric patients in intensive care units and is highly prevalent among deceased children. However, the study did not analyze the independent association between hyperkalemia and adverse outcomes [17]. Our study contributes to the limited literature on the relationship between serum potassium levels and outcomes in pediatric patients. Notably, we found that potassium levels of ≥4.7 mmol/L, which are lower than the upper limit of the normal range (5.5 mmol/L), were independently associated with an increased risk of 1-year readmission in this population. Furthermore, our analysis identified an optimal potassium level of 4.1 mmol/L to be associated with the lowest risk of readmission. These findings underscore the importance of maintaining serum potassium levels within a narrower range than the conventional reference range to improve outcomes in pediatric patients with HF.

The mechanisms underlying the association between hyperkalemia and increased risk of readmission in pediatric patients with HF are likely multifactorial. Hyperkalemia is frequently observed in patients with HF, renal insufficiency, and acid–base imbalances and in those receiving RAASi [18,19]. In our study, patients in the high-potassium group exhibited poorer kidney and cardiac function indicators and a higher prevalence of prior HF history, suggesting that hyperkalemia may serve as a marker for advanced disease states. However, the independent association between potassium levels ≥4.7 mmol/L and 1-year readmission risk persisted after adjusting for these factors, indicating that hyperkalemia itself may contribute to adverse outcomes. In addition to being a marker of disease severity, hyperkalemia can directly affect cardiac function. Elevated serum potassium levels alter the resting membrane potential of cardiomyocytes, leading to impaired conduction velocity and decreased excitability [20,21]. This can result in arrhythmias, heart blocks, and reduced myocardial contractility [22]. These direct effects of hyperkalemia on cardiac function may further contribute to the increased risk of adverse outcomes observed in pediatric patients with HF and elevated serum potassium levels, independent of the underlying disease severity.

Interestingly, our study did not reveal a significant association between hypokalemia and increased readmission risk, in contrast to previous studies on adult HF populations, regardless of the inclusion of covariates. Similar findings were recently reported by Liu et al. and Shang et al., indicating that only higher potassium levels are associated with an increased incidence of adverse outcome events in patients with HF [15,23]. This discrepancy may be due to the smaller sample size of the low-potassium group in our study or the fact that pediatric patients with hypokalemia at baseline may have received potassium supplementation during hospitalization, mitigating the potential adverse effects of low potassium levels and emphasizing the importance of promptly addressing and correcting hypokalemia in pediatric patients with HF.

The clinical implications of our findings are two-fold. First, our study underscores the importance of closely monitoring and managing serum potassium levels in pediatric patients with HF, aiming for a narrower optimal range than the conventional reference range. Second, our results suggest that hyperkalemia may serve as a prognostic marker in pediatric HF, identifying patients at a higher risk of readmission who may benefit from more intensive follow-up and management.

This study has several limitations. First, this was a single-center observational study; therefore, the potential influence of residual confounding factors may not be completely eliminated. Therefore, the generalizability of these findings to other populations should be interpreted with caution. Second, factors such as medical history and prior medication use, particularly diuretics, can influence baseline potassium levels. However, the retrospective nature of this study precluded the collection of detailed information on patients’ medication use prior to hospitalization. The absence of data on pre-hospitalization diuretic use is a limitation that should be addressed in future prospective studies. Third, only baseline potassium values were incorporated, and no assessment was conducted in terms of the influence of therapeutic interventions on these levels. Therefore, the effects of adjusting serum potassium levels on patient prognosis could not be definitively confirmed.

## 5. Conclusions

Our study demonstrated a J-shaped relationship between baseline serum potassium levels and 1-year readmission risk in pediatric patients with HF, with an optimal potassium level of 4.1 mmol/L. A potassium level of ≥4.7 mmol/L was independently associated with increased readmission risk, highlighting the importance of maintaining potassium levels within a narrower optimal range in this population.

## Figures and Tables

**Figure 1 children-11-00725-f001:**
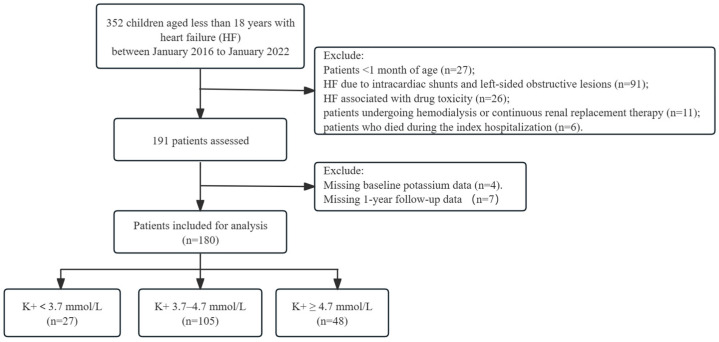
Flowchart of the patient selection process.

**Figure 2 children-11-00725-f002:**
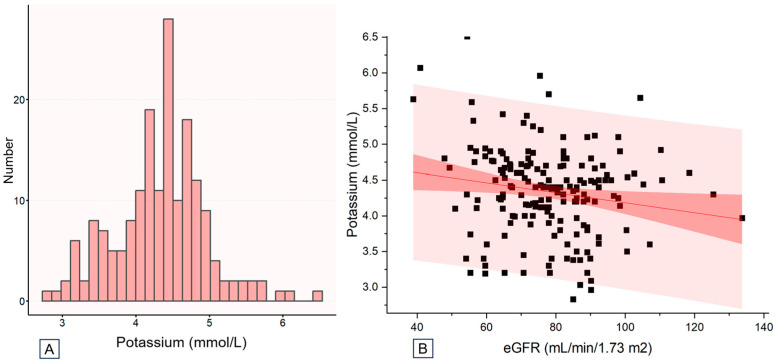
Distribution of baseline potassium values of the study population. (**A**) Overall distribution of baseline potassium. (**B**) Distribution of baseline potassium by estimated glomerular filtration rate. Note: No variables were adjusted. Abbreviations: eGFR, estimated glomerular filtration rate.

**Figure 3 children-11-00725-f003:**
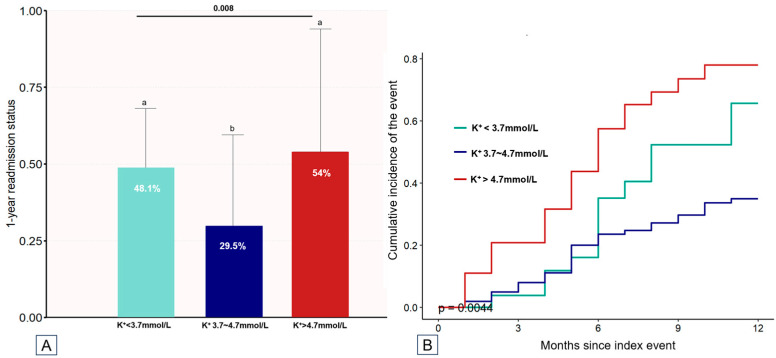
Baseline potassium levels and 1-year readmission in pediatric patients with HF. (**A**) The incidence of 1-year readmission. Note: Bars sharing the same letter (a or b) are not significantly different from each other, while bars with different letters indicate significant differences (*p* < 0.05). (**B**) cumulative incidence of 1-year readmission.

**Figure 4 children-11-00725-f004:**
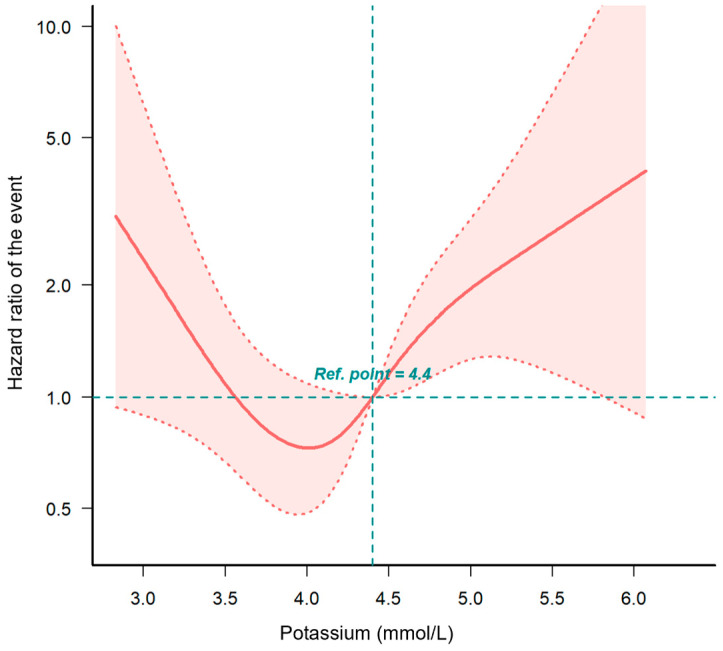
Risk of 1-year readmission by baseline potassium level. Notes: The value on the horizontal axis corresponding to a hazard ratio (HR) of 0 represents the reference level of 4.4 mmol/L for baseline potassium, indicating the risk of 1-year readmission in the absence of the studied potassium levels. Adjusted for all covariates in Table 2 Model 3.

**Table 1 children-11-00725-t001:** Baseline characteristics and outcome of the study patients.

Variables	Total (*n* = 180)	Baseline Serum Potassium (mmol/L)			*p* Value
		<3.7 (*n* = 27)	3.7–4.7 (*n* = 105)	≥4.7 (*n* = 48)	
Age, year	7.0 ± 5.6	6.9 ± 5.4	6.6 ± 5.4	7.8 ± 6.1	0.488
Sex, male, *n* (%)	106 (58.9)	16 (59.3)	59 (56.2)	31 (64.6)	0.691
Ethnicity, *n* (%)					0.993
Han	79 (43.9)	13 (48.1)	46 (43.8)	20 (41.7)	
Zhuang	87 (48.3)	13 (48.1)	50 (47.6)	24 (50)	
Miao	8 (4.4)	1 (3.7)	5 (4.8)	2 (4.2)	
Other	6 (3.3)	0 (0)	4 (3.8)	2 (4.2)	
Preexisting HF, *n* (%)	31 (17.2)	3 (11.1)	14 (13.3)	14 (29.2)	0.043
NYHA/Ross classification, *n* (%)					0.046
II	76 (42.2)	13 (48.1)	50 (47.6)	13 (27.1)	
III/IV	104 (57.8)	14 (51.9)	55 (52.4)	35 (72.9)	
Hospital length of stay, days	11.4 ± 6.8	11.6 ± 6.5	10.7 ± 6.3	12.7 ± 7.9	0.252
Etiology of HF, *n* (%)	31 (17.2)	3 (11.1)	14 (13.3)	14 (29.2)	0.951
Dilated cardiomyopathy	121 (67.2)	19 (70.4)	70 (66.7)	32 (66.7)	
Myocarditis	34 (18.9)	4 (14.8)	20 (19)	10 (20.8)	
Anomalous origin of coronary artery	7 (3.9)	0 (0)	6 (5.7)	1 (2.1)	
Kawasaki disease	11 (6.1)	2 (7.4)	6 (5.7)	3 (6.2)	
Hypertensive disease	4 (2.2)	1 (3.7)	2 (1.9)	1 (2.1)	
Other	3 (1.7)	1 (3.7)	1 (1)	1 (2.1)	
BUN, mmol/L	6.1 ± 3.3	6.3 ± 2.6	5.7 ± 2.6	7.0 ± 4.6	0.054
SCr, µmol/L	46.3 ± 23.9	47.3 ± 20.6	43.3 ± 21.6	52.5 ± 29.2	0.084
eGFR, mL/min/1.73 m^2^	77.8 ± 15.1	79.1 ± 14.3	80.0 ± 14.5	72.3 ± 15.6	0.012
SUA, µmol/L	375.1 ± 157.8	389.1 ± 158.2	369.5 ± 153.8	379.7 ± 168.6	0.826
ALB, g/L	39.8 ± 4.3	40.2 ± 4.0	39.7 ± 4.7	39.8 ± 3.7	0.894
HB, g/L	120.4 ± 20.2	118.5 ± 19.4	120.7 ± 18.7	121.0 ± 23.8	0.863
Serum sodium, mmol/L	135.4 ± 4.8	136.0 ± 5.2	135.7 ± 4.3	134.5 ± 5.5	0.249
LVEF, %	32.0 ± 10.0	32.4 ± 8.7	32.0 ± 10.1	31.9 ± 10.7	0.029
Discharge medication					
Digoxin, *n* (%)	109 (60.6)	18 (66.7)	59 (56.2)	32 (66.7)	0.336
Captopril, *n* (%)	82 (45.6)	14 (51.9)	40 (38.1)	28 (58.3)	0.051
Hydrochlorothiazide, *n* (%)	113 (62.8)	15 (55.6)	67 (63.8)	31 (64.6)	0.699
Spironolactone, *n* (%)	129 (71.7)	20 (74.1)	72 (68.6)	37 (77.1)	0.531
Metoprolol, *n* (%)	55 (30.6)	10 (37)	32 (30.5)	13 (27.1)	0.668
Outcome					
1 or more 1-year readmissions for HF, *n* (%)	70 (38.9)	13 (48.1)	31 (29.5)	26 (54.2)	0.008
≥2 admissions for HF, *n* (%)	32 (17.8)	5 (18.5)	11 (10.5)	16 (33.3)	0.005

Notes: Data are presented as mean ± SD for normally distributed variables, median (Q1, Q3) for skewed variables, and *n* (%) for categorical variables. *p* < 0.05 are shown in bold. Abbreviations: HF, heart failure; NYHA, New York Heart Association; BUN, blood urea nitrogen; SCr, serum creatinine; eGFR, estimated glomerular filtration rate; SUA, serum uric acid; ALB, serum albumin; HB, hemoglobin; LVEF, left ventricular ejection fraction.

**Table 2 children-11-00725-t002:** Multivariate Cox regression analyses for baseline potassium on the risk of 1-year readmission for HF.

Variable	Unadjusted Model		Model 1		Model 2		Model 3	
	HR_95% CI	*p* Value	HR_95% CI	*p* Value	HR_95% CI	*p* Value	HR_95% CI	*p* Value
Potassium per 1 mmol/L increase	1.85 (1.19−2.87)	0.006	1.83 (1.19−2.82)	0.006	1.36 (0.9−2.05)	0.141	1.3 (0.85−2.01)	0.227
Potassium < 3.7 mmol/L	1.71 (0.9−3.28)	0.103	1.63 (0.84−3.14)	0.146	1.55 (0.81−2.97)	0.19	1.72 (0.89−3.32)	0.109
Potassium 3.7−4.7 mmol/L	1 (Ref)		1 (Ref)		1 (Ref)		1 (Ref)	
Potassium ≥ 4.7 mmol/L	2.34 (1.39−3.94)	0.001	2.02 (1.16−3.53)	0.013	2.03 (1.17−3.51)	0.012	1.91 (1.09−3.35)	0.024

Notes: Model 1: adjusted for age + sex; Model 2: adjusted for Model 1 BUN + SCr + eGFR +SUA; Model 3: adjusted for Model 2 + LVEF + NYHA/Ross classification + ALB + serum sodium. Abbreviations: HF, heart failure; NYHA, New York Heart Association; BUN, blood urea nitrogen; SCr, serum creatinine; eGFR, estimated glomerular filtration rate; SUA, serum uric acid; ALB, serum albumin; LVEF, left ventricular ejection fractions.

**Table 3 children-11-00725-t003:** Univariate and multivariable Cox regression analyses evaluating the risk predictors of 1-year readmission.

Variables	Univariate Cox Analyses HR (95% CI)	*p* Value	Multivariable Cox Analyses HR (95% CI)	*p* Value
Potassium ˂ 3.7 mmol/L	1.71 (0.9−3.28)	0.103	-	
Potassium 3.7−4.7 mmol/L	Ref			
Potassium ≥ 4.7 mmol/L	2.34 (1.39−3.94)	0.001	1.87 (1.08−3.26)	0.026
Age, year	1.05 (1−1.09)	0.032	0.98 (0.93−1.04)	0.512
Sex, male	0.79 (0.48−1.28)	0.327	-	
Hospital length of stay, day	1.03 (1−1.07)	0.051	-	
SCr, µmol/L	1.01 (1.01−1.02)	0.002	1 (0.99−1.01)	0.906
BUN, mmol/L	1.16 (1.1−1.23)	<0.001	1.09 (1.01−1.18)	0.029
eGFR, per 10 mL/min/1.73 m^2^ increase	0.79 (0.66−0.94)	0.006	0.93 (0.77−1.13)	0.467
SUA, per 10 µmol/L increase	1.02 (1−1.03)	0.041	1.01 (0.99−1.02)	0.425
ALB, g/L	0.94 (0.89−0.99)	0.019	0.95 (0.89−1.01)	0.077
Serum sodium, mmol/L	0.94 (0.9−0.98)	0.013	0.99 (0.94−1.05)	0.757
HB, per 10 g/L increase	1.02 (0.91−1.14)	0.776	-	
LVEF, %	0.98 (0.95−1)	0.06	-	
Preexisting HF ≥ 6 months	1.7 (0.97−2.97)	0.077	-	
NYHA/Ross classification, III/IV	2.39 (1.41−4.05)	0.001	1.94 (1.09−3.46)	0.025
Digoxin at discharge	0.79 (0.43−1.46)	0.455	-	
Captopril at discharge	1.34 (0.73−2.45)	0.34	-	
Hydrochlorothiazide at discharge	0.89 (0.55−1.44)	0.645		
Spironolactone at discharge	1.43 (0.82−2.51)	0.191		
Metoprolol at discharge	0.61 (0.31−1.19)	0.147	-	

Notes: *p* < 0.05 were shown in bold in Multivariable Cox analyses. Abbreviations: HF, heart failure; NYHA, New York Heart Association; BUN, blood urea nitrogen; SCr, serum creatinine; eGFR, estimated glomerular filtration rate; SUA, serum uric acid; ALB, serum albumin; HB, hemoglobin; LVEF, left ventricular ejection fraction.

## Data Availability

All raw data generated during the course of this study have been comprehensively incorporated within this published article and its Appendix A.

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
