# Peer review of "Relationship between Baseline Serum Potassium and 1-Year Readmission in Pediatric Patients with Heart Failure: A Retrospective Cohort Study"

_children, 2024, doi:10.3390/children11060725_

Round 1
Reviewer 1 Report
Comments and Suggestions for Authors
The role of the kidney is essential in maintaining overall homeostasis of ion concentrations in the blood. Concentration gradient of potassium across the cell membrane is a determinant of the membrane potential of cells, small variations in serum potassium level may lead to severe heart dysfunction. Deficiencies in potassium intake have been associated with adverse cardiovascular events. The maturation process of the kidney is still going on until the age of five or six years old. These may represent some bias. The topic may be of interest.
Comments on the Quality of English LanguageEnglish requires minor editing
Author Response
Dear Editor and Reviewers,
We would like to thank you for your feedback regarding the language and editing of our manuscript. We appreciate your attention to detail and the opportunity to improve the clarity and readability of our work.
As per your suggestion, we have carefully reviewed the manuscript and made necessary revisions to enhance the quality of the English language used throughout the text. We have focused on improving grammar, sentence structure, and overall coherence to ensure that our findings are effectively communicated to the reader.
To assist us in this process, we have sought the help of a native English speaker with expertise in scientific writing to review and edit our manuscript. We believe that these efforts have significantly elevated the linguistic quality of our work.
We thank you once again for your valuable input and hope that the revised manuscript meets your expectations in terms of language and clarity.
Please let us know if you have any further questions or require additional information.
Sincerely,
Yong Han
Reviewer 2 Report
Comments and Suggestions for Authors
Material and methods
It would be important to specify what kind of diuretics were given in patients with heart failure (loop diuretics or potassium sparing diuretics or association of them; dose of diuretics). The value of potassium could depend on the type of diuretics used and dose, so correlation between the dose, type and association of drugs should increase the value of this work.
Results
Page 4, table 1
Please specify the value of LVEF for the group of patients having K value more than 4.7mmol/L.
Please specify if there is a correlation between the dose, type and association of diuretics used in the group of children with heart failure.
Please specify the number of readmissions in the first year after the diagnosis of heart failure and the correlation with the potassium level.
Author Response
Dear Editor and Reviewers,
Thank you for your thoughtful comments and suggestions regarding our manuscript. We appreciate the opportunity to address the points you have raised and provide further clarification.
1. We would like to thank you for raising an important question about the potential influence of pre-hospitalization medication use on baseline potassium levels. This is indeed a crucial aspect to consider when interpreting our findings. We would like to emphasize that the medications presented in Table 1 are the discharge medications prescribed to patients upon leaving the hospital. The potassium levels we collected and analyzed were the baseline potassium levels, which were measured during the patients' first blood test upon hospital admission. As you correctly pointed out, factors such as medical history and prior medication use (particularly diuretics) can influence baseline potassium levels. Due to the retrospective nature of our study, we were unable to obtain detailed information on the patients' medication use prior to hospitalization. However, we were able to extract data on the patients' history of heart failure from their medical records, and we have included this information in our analysis of factors potentially affecting baseline potassium levels. We acknowledge that the lack of data on pre-hospitalization diuretic use is a limitation of our study, and we have addressed this in the limitations section of the manuscript. We agree that this is an important aspect to consider, and we plan to explore this further in future prospective studies. Additionally, our study included an analysis of the medications prescribed at discharge, such as diuretics and digoxin, and we have performed a detailed analysis of these specific medications to provide a more comprehensive understanding of their potential impact on the study outcomes (changes marked in red).
2. We have added the LVEF value for the K>4.7 group in Table 1 (marked in red).
3. We have supplemented Table 1 with information on the relationship between different potassium levels and the occurrence of 1 or more readmissions and more than 2 readmissions. Additionally, we have provided further details in the Results section 3.3, stating, "Among the study population, 38.9% experienced at least 1 readmission for HF within 1 year, with 17.8% having 2 or more readmissions. The mean number of readmissions per patient readmitted at least once was 2.1±1.3 (median, 1.2; IQR, 1–3)." (marked in red).
We hope that these clarifications and additions adequately address your concerns and improve the quality of our manuscript. Please let us know if you have any further questions or require additional information.
Sincerely,
Yong Han
Reviewer 3 Report
Comments and Suggestions for Authors
I thank the authors for the opportunity I have had to read this interesting manuscript, and I congratulate them for the work they have done.
In order to improve the manuscript, I allow myself to make some comments.
In the results section, the aurors indicate the age using the median and in Table 1 using the mean. They should unify the result (mean or median). If the variable does not follow normality, it is imperative to use the median and therefore a non-parametric test for comparisons.
At the bottom of Table 1 it is indicated that there are data that are presented as median and interquartile range, however, no variable that is presented in this way is targeted.
Numerical data presented in tables should not be repeated in the text (redundant data). Therefore, the entire paragraph that begins on line 192 should be deleted.
The last sentence of the conclusions does not follow from the results and should be eliminated (it is an appropriate assessment for the conclusions section).
Author Response
Dear Editor and Reviewers,
We would like to express our sincere gratitude for your thorough review and invaluable suggestions regarding our manuscript. The points you have raised are extremely pertinent and have significantly contributed to improving the clarity, consistency, and overall quality of our work.
1. We have unified the presentation of age in the results section and Table 1 using the median and interquartile range (IQR), as the variable does not follow a normal distribution. Accordingly, we have also updated the statistical tests for comparisons to non-parametric tests (changes marked in blue).
2. We apologize for the oversight in Table 1. We have now clearly indicated the variables presented as median and interquartile range (changes marked in blue).
3. As per your recommendation, we have removed the redundant numerical data from the text that was already presented in Table 1. The entire paragraph beginning on line 192 has been deleted (changes marked in blue).
4. We agree with your assessment regarding the last sentence of the conclusions. As it does not directly follow from the results, we have removed this sentence from the conclusions section (changes marked in blue).
We believe that incorporating your suggestions has greatly enhanced the readability and scientific rigor of our manuscript. We are grateful for your expertise and the time you have invested in reviewing our work.
Please let us know if you have any further questions or require additional information.
Sincerely,
Yong Han
Round 2
Reviewer 1 Report
Comments and Suggestions for Authors
The text presents some interesting aspects of the management of heart failure in children.
Comments on the Quality of English LanguageThe English language requires minor editing.